# Properties of Spherically Symmetric Black Holes in the Generalized Brans–Dicke Modified Gravitational Theory

**DOI:** 10.3390/e25050814

**Published:** 2023-05-18

**Authors:** Mou Xu, Jianbo Lu, Shining Yang, Hongnan Jiang

**Affiliations:** Department of Physics, Liaoning Normal University, Dalian 116029, China; xumou@lnnu.edu.cn (M.X.);

**Keywords:** generalized Brans–Dicke theory, corrected entropy of black hole, stability of particles’ circular orbits, modified gravity theory

## Abstract

The many problems faced by the theory of general relativity (GR) have always motivated us to explore the modified theory of GR. Considering the importance of studying the black hole (BH) entropy and its correction in gravity physics, we study the correction of thermodynamic entropy for a kind of spherically symmetric black hole under the generalized Brans–Dicke (GBD) theory of modified gravity. We derive and calculate the entropy and heat capacity. It is found that when the value of event horizon radius $r_{+}$ is small, the effect of the entropy-correction term on the entropy is very obvious, while for larger values $r_{+}$, the contribution of the correction term on entropy can be almost ignored. In addition, we can observe that as the radius of the event horizon increases, the heat capacity of BH in GBD theory will change from a negative value to a positive value, indicating that there is a phase transition in black holes. Given that studying the structure of geodesic lines is important for exploring the physical characteristics of a strong gravitational field, we also investigate the stability of particles’ circular orbits in static spherically symmetric BHs within the framework of GBD theory. Concretely, we analyze the dependence of the innermost stable circular orbit on model parameters. In addition, the geodesic deviation equation is also applied to investigate the stable circular orbit of particles in GBD theory. The conditions for the stability of the BH solution and the limited range of radial coordinates required to achieve stable circular orbit motion are given. Finally, we show the locations of stable circular orbits, and obtain the angular velocity, specific energy, and angular momentum of the particles which move in circular orbits.

## 1. Introduction

Einstein established the famous theory of special relativity in 1905, which laid the foundation for the relativistic view of spacetime. Based on his deep thinking about special relativity, Einstein established the theory of general relativity (GR) in 1915. GR is considered one of the most elegant theories in the field of physics, and has received significant experimental support. For example: (1) general relativity passed rigorous tests within the solar system; (2) its applications in cosmology have also been extremely successful; (3) the detection of gravitational waves by the advanced Laser Interferometer Gravitational-Wave Observatory (LIGO) in 2015 provided strong support for the theoretical predictions made by GR 100 years ago [1]; (4) the fact that the Event Horizon Telescope (EHT) [2] discovered the existence of supermassive black holes in the center of the Milky Way galaxy in 2019 greatly enhanced interest and confidence in the research of general relativity theory. However, it is also evident that many problems and challenges remain in the study of gravitational physics, such as the problem of gravitational quantization, the nature of dark energy, the problem of inflationary universe, and the problem of space–time singularity. To address these issues, many modified gravity theories have been developed and discussed. For example, the Ricci scalar *R* in the Einstein–Hilbert action can be generalized to an arbitrary function f(R), termed as the f(R) theory [3,4]; considering the teleparallel equivalent of GR, a torsion-based $f(\tilde{T})$ gravity was given which is the simplest nonlinear torsional modified theory [5,6] with the advantage that its field equations have up to second-order derivatives; an extended theory of f(R), f(R,T) theory [7] was proposed by introducing the coupling between the Ricci scalar *R* and the trace of energy-momentum tensor *T*; in $f(\tilde{G})$ theory [8], $f(\tilde{G})$ is an arbitrary function of the Gauss–Bonnet invariant $\tilde{G}$ that can be constructed by the Ricci scalar, Ricci tensor, and Riemann tensors; amongst the theories alternative to GR, the simplest theory is to add a scalar field to GR, known as the scalar–tensor (ST) theory [9,10,11,12,13,14], the most straightforward and natural method of modifying GR; Scalar–tensor–vector (STV) gravity theory was proposed by Moffat [15]. STV theory includes, in addition to the metric tensor, three scalar fields (related to the Newtonian gravitational constant, the coupling function of field, and the rest mass of the field) and a vector field (associated with a fifth force charge) [15,16]. The exploration of modified theories of gravity remain a hot topic in the current research of gravitational physics. As one type of modified gravity theory, the generalized Brans–Dicke (GBD) theory has been studied in cosmology, gravitational wave physics, and other fields [17,18,19]. In this paper, we will explore some issues in the framework of GBD theory.

Black holes (BHs) are one of the important predictions of general relativity. After years of observation and exploration, researchers finally captured the first image of a BH in 2019 through the Event Horizon Telescope. In fact, what the EHT observed was a black hole shadow [2], the lensed image at infinity of the photon sphere [20,21]. The circular photon orbit is intimately related to the black hole shadow and can be closely associated with the spacetime geometric structure. Therefore, it serves as a robust tool for estimating black hole parameters [22,23,24,25,26] and for testing GR or its alternatives [27,28,29,30,31,32,33]. Consequently, to probe the nature of black holes, the shadows of different black holes have been studied under various modified theories of gravity [20,26,34,35,36,37].

For many years, the research on BH physics has attracted the attention of physicists and astronomers. Different types of black holes, such as static BHs [38], dynamic BHs [39], spherically symmetric BHs [40], axially symmetric BHs [41], and exotic BHs [42], have been intensively discussed. The study of the thermodynamic laws of BH areas suggests that black holes, as special celestial bodies, seem to have thermal properties. It is well known that the thermal properties of BHs under different types have been widely explored. The research on the corrected BH entropy is a hot topic in BH thermodynamics, and some studies on the correction of black hole entropy under modified gravity theory have been investigated, such as conformal field theory [43,44], string theory [45,46], and others. This article mainly explores the relevant properties of thermodynamic correction entropy of static spherically symmetric BHs within the framework of GBD modified gravity theory. Additionally, the stability analysis of particles’ circular orbits around a BH is an important research topic in the field of gravitational physics, which plays a vital role in exploring the properties of black holes and gravity. Under different contexts of BH, people have conducted extensive research on the stability of black holes through the application of geodesic deviation equation, such as charged black holes in $f(\tilde{T})$ theory [47], rotating (anti-) de-Sitter black holes in f(R) theory [48], and non-trivial black holes [49,50,51,52,53]. In this paper, we also discuss the stability of particles’ circular orbits around a black hole in the GBD gravity theory.

The structure of our paper is as follows. The Section 1 is an introduction. In the Section 2 of this paper, we focus on the correction of thermodynamic entropy of a static spherically symmetric BH within the framework of GBD modified gravity theory. We analyze the stability of particles’ circular orbits in black holes in the Section 3. The Section 4 is the conclusion of this article.

## 2. Entropy Correction of Black Hole Thermodynamics in GBD Modified Gravity Theory

We first briefly introduce the GBD theory, whose action is written as [17]:(1)$$S=\int d^{4}x\sqrt{-g}\left[\phi f\left(R\right)-\frac{\omega }{\phi }\partial_{\mu }\phi \partial^{\mu }\phi +\frac{16\pi }{c^{4}}L_{m}\left(g_{\mu \nu },\psi_{m}\right)\right].$$

Here $g_{\mu \nu }$ denotes the metric, *R* is the Ricci scalar, and f(R) represents an arbitrary function relative to *R*. ϕ denotes the Brans–Dicke scalar field, ω is the coupling constant, $L_{m}$ represents the Lagrange density of matter field $\psi_{m}$. By applying the variational principle, the gravitational field equation and scalar field equation of GBD theory can be derived as follows:(2)$$\begin{array}{c} \phi \left[f_{R}R_{\mu \nu }-\frac{1}{2}f\left(R\right)g_{\mu \nu }\right]-\left(\nabla_{\mu }\nabla_{\nu }-g_{\mu \nu }□\right)\left(\phi f_{R}\right)+\frac{1}{2}\frac{\omega }{\phi }g_{\mu \nu }\partial_{\sigma }\partial^{\sigma }\phi -\frac{\omega }{\phi }\partial_{\mu }\phi \partial_{\nu }\phi =8\pi T_{\mu \nu }. \end{array}$$
(3)$$\begin{array}{c} f\left(R\right)+2\omega \frac{□\phi }{\phi }-\frac{\omega }{\phi^{2}}\partial_{\mu }\phi \partial^{\mu }\phi =0. \end{array}$$

We consider a specific parametric model [54,55]: $\phi \left(r\right)=\phi_{0}r^{-a}$ and $f\left(R\right)=R+\beta R^{\left(-n\right)}$. Under the background of the spherically symmetric space–time line elements:(4)$$\begin{array}{c} d\tau^{2}=-B\left(r\right)dt^{2}+A\left(r\right)dr^{2}+r^{2}d\theta^{2}+r^{2}sin\theta^{2}d\varphi^{2}. \end{array}$$
with $A\left(r\right)=B{\left(r\right)}^{-1}$, the solution of the field equation can be calculated when n≥−1
(5)$$\begin{array}{c} B\left(r\right)=1+\frac{C_{1}}{r}+\frac{C_{2}}{r^{2}}. \end{array}$$
where $C_{1}$ and $C_{2}$ are two constant parameters. The speed of light *c* and the current value of the gravitational constant *G* have been taken as the geometrized units in this paper, c=G=1. Obviously, expression (Equation 5) is similar in form to the Reissner–Nordstrom (RN) solution for charged celestial bodies in Einstein’s general relativity (if the integration constants are set to $C_{1}=-2M$, $C_{2}=\frac{Q^{2}}{4\pi }$, then the two solutions are the same in form). Therefore, in the subsequent discussion, we consider that the values of these two parameters satisfy $C_{1}\leq 0$ and $C_{2}\geq 0$. Given *Q* is the charge parameter in RN solution, it seems that there exists a similar scalar charge in GBD theory. In addition, for $C_{1}=-2M$ and $C_{2}=0$, Equation (Equation 5) is reduced to the Schwarzschild form in GR. It indicates that the parameter $C_{1}$ represents the mass of BH.

The infinite red-shift surface is defined by $g^{\mu \nu }\frac{\partial f}{\partial x^{\mu }}\frac{\partial f}{\partial x^{\nu }}=0$. According the solution (Equation 5), we obtain the infinite red-shift surface for the BH in GBD: $r_{\pm }=\frac{-C_{1}\pm \sqrt{C_{1}^{2}-4C_{2}}}{2}$ with $C_{1}^{2}-4C_{2}\geq 0$. For the static spherically symmetry solution of BH in GBD theory, the event horizon $r_{h}=r_{+}$ is overlapped with the infinite red-shift surface. In addition, the properties of the spacetime metric of the RN black hole with electric charge *Q* and its generalized forms can be seen in [36,56,57]. Obviously, the properties of the BH spacetime metric under GBD theory differ from those of the generalized RN black holes, e.g., the RN black hole with the cosmological constant [57] and the regular charged black hole [36].

The study of black hole thermodynamics and entropy correction has always been an important issue in the gravitational physics. Next, we investigate the corrected entropy of spherically symmetric BH in the framework of GBD theory. Consider the partition function in the form [58]:(6)$$\begin{array}{c} Z\left(\beta \right)=\int_{0}^{\infty }\rho \left(E\right)e^{\left(-\beta E\right)}dE. \end{array}$$
where $T=\frac{1}{\beta }$ is the temperature in units of the Boltzmann constant $\kappa_{B}$. ρ(E) is the density of state, which can be given from (Equation 6) by performing the inverse Laplace transform (keeping *E* fixed) [59,60]:(7)$$\begin{array}{c} \rho \left(E\right)=\frac{1}{2\pi i}\int_{c-i\infty }^{c+i\infty }Z\left(\beta \right)e^{\beta E}d\beta =\frac{1}{2\pi i}\int_{c-i\infty }^{c+i\infty }e^{S(\beta )}d\beta . \end{array}$$

In Equation (Equation 7), the exact entropy as a function of temperature (not just at equilibrium) can be written as [61]
(8)$$\begin{array}{c} S(\beta )=lnZ(\beta )+\beta E. \end{array}$$

It is formally defined as the sum of entropies of subsystems of the thermodynamic system, which are small enough to be themselves in equilibrium [61]. Considering the method of steepest descent around the saddle point: $\beta_{0}=\frac{1}{T_{0}}$, we have $S_{0}^{{}^{\prime }}\equiv \frac{\partial S(\beta )}{\partial \beta }{|}_{\beta =\beta_{0}}=0$ with $T_{0}$ being the equilibrium temperature. On the other hand, we can receive the following form of entropy function by expanding it about $\beta =\beta_{0}$,
(9)$$\begin{array}{c} S\left(\beta \right)=S\left(\beta_{0}\right)+S^{{}^{\prime }}\left(\beta_{0}\right)\left(\beta -\beta_{0}\right)+\frac{S^{{}^{\prime \prime }}\left(\beta_{0}\right)}{2}{\left(\beta -\beta_{0}\right)}^{2}+\ldots \end{array}$$

Then Equation (Equation 9) can be simplified as
(10)$$\begin{array}{c} S\left(\beta \right)=S_{0}+\frac{1}{2}{\left(\beta -\beta_{0}\right)}^{2}S^{{}^{\prime \prime }}\left(\beta_{0}\right)+\ldots \end{array}$$
with
(11)$$\begin{array}{c} S_{0}\equiv S\left(\beta_{0}\right),S_{0}^{{}^{\prime \prime }}\equiv \frac{\partial^{2}S\left(\beta \right)}{\partial \beta^{2}}{|}_{\beta =\beta_{0}}. \end{array}$$
where S(β) represents the entropy at any temperature, and $S_{0}$ is the entropy calculated by the Beckenstein–Hawking area law. As can be seen from Equation (Equation 9), the correction of entropy is only described by the term of $S_{0}^{{}^{\prime \prime }}$. Using Equation (Equation 8), it can be deduced that
(12)$$\begin{array}{c} S^{{}^{\prime }}\left(\beta \right)=\frac{1}{Z(\beta )}\frac{\partial Z(\beta )}{\partial \beta }+E. \end{array}$$
(13)$$\begin{array}{c} S^{{}^{\prime \prime }}\left(\beta \right)=-\frac{1}{Z^{2}\left(\beta \right)}{\left(\frac{\partial Z(\beta )}{\partial \beta }\right)}^{2}+\frac{1}{Z(\beta )}\frac{\partial^{2}Z\left(\beta \right)}{\partial \beta^{2}}. \end{array}$$

Consider the energy of a canonical ensemble [61]:(14)$$\begin{array}{c} E=\bar{E}=-\frac{\partial }{\partial \beta }lnZ=-\frac{1}{Z}\frac{\partial Z}{\partial \beta }. \end{array}$$
(15)$$\begin{array}{c} \frac{\partial \bar{E}}{\partial \beta }=\frac{1}{Z^{2}\left(\beta \right)}{\left(\frac{\partial Z(\beta )}{\partial \beta }\right)}^{2}-\frac{1}{Z(\beta )}\frac{\partial^{2}Z\left(\beta \right)}{\partial \beta^{2}}. \end{array}$$

Then we have:(16)$$\begin{array}{c} \frac{\partial \bar{E}}{\partial \beta }=-\left[<E^{2}>-<E{>}^{2}\right]=-\kappa_{B}T^{2}C. \end{array}$$
here *C* is the heat capacity. So, we finally obtain
(17)$$\begin{array}{c} S_{0}^{{}^{\prime \prime }}=T^{2}C. \end{array}$$

Applying it to a thermodynamic system of BH, the saddle point is represented as $\beta_{0}=\frac{1}{T_{H}}$ by taking Hawking temperature $T_{H}$ instead of *T*. Following the method used in reference [62], the entropy can be rewritten by introducing a parameter γ to track the correction term
(18)$$\begin{array}{c} S=S_{0}-\gamma \left(CT^{2}\right). \end{array}$$

For GBD modified gravity theory, $S_{0}$ can be derived as follows
(19)$$\begin{array}{c} S_{0}=\pi r_{+}^{2}\varphi_{0}\frac{1}{r_{+}^{\alpha }}\left(1-n\frac{\beta }{R^{n+1}}\right). \end{array}$$

Comparing the spherically symmetric BH solution (Equation 5) under this theory with the Schwarzschild BH solution in the GR theory, the integration constant $C_{1}$ can be easily expressed as $C_{1}=-2M$, namely
(20)$$\begin{array}{c} B\left(r\right)=1-\frac{2M}{r}+\frac{C_{2}}{r^{2}}. \end{array}$$

Considering relation: $R\left(r\right)=\frac{2-2B\left(r\right)-4rB^{{}^{\prime }}\left(r\right)-r^{2}B^{{}^{\prime \prime }}\left(r\right)}{r^{2}}$ obtained by line elements (Equation 4) and using Equations (Equation 19) and (Equation 20), we can derive:(21)$$\begin{array}{c} S_{0}=\pi r_{+}^{2-\alpha }, \end{array}$$
letting $\varphi_{0}=1$. Then expression of the event horizon radius can be written as
(22)$$\begin{array}{c} r_{+}=\pi^{-\frac{1}{2-\alpha }}S_{0}^{\frac{1}{2-\alpha }}. \end{array}$$

Obviously, the relationship between the BH mass and the event horizon radius in GBD theory meets
(23)$$\begin{array}{c} M=\frac{r_{+}}{2}+\frac{C_{2}}{2r_{+}}. \end{array}$$

Using expression (Equation 23), we draw the picture of the black hole mass relative to the event horizon radius for different parameter values $C_{2}$, as shown in Figure 1. From Figure 1, we observe that for the case $C_{2}=15$, the black hole mass reaches the minimum value M≈3.87 when the event horizon radius is $r_{+}\approx 3.87$.

Combining Equations (Equation 21) and (Equation 23), we obtain
(24)$$\begin{array}{c} M=\frac{1}{2}+\pi^{\frac{1}{2-\alpha }}S_{0}^{-\frac{1}{2-\alpha }}\left(\pi^{\frac{1}{2-\alpha }}S_{0}^{\frac{2}{2-\alpha }}+C_{2}\right). \end{array}$$

Then we can obtain the concrete expressions of temperature and heat capacity for the GBD black hole as follows:(25)$$\begin{array}{c} T=\frac{\partial M}{\partial S_{0}}=\frac{\pi^{-\frac{1}{2-\alpha }}S_{0}^{\frac{-1+\alpha }{2-\alpha }}}{2-\alpha }-\frac{\pi^{\frac{1}{2-\alpha }}S_{0}^{\frac{\alpha -3}{2-\alpha }}\left(\pi^{-\frac{1}{2-\alpha }}S_{0}^{\frac{2}{2-\alpha }}+C_{2}\right)}{2*(2-\alpha )} \end{array}$$
(26)$$\begin{array}{c} C=\frac{\partial M}{\partial T}=T\frac{\partial S_{0}}{\partial T}=-\frac{S_{0}*\left(-2+\alpha \right)\left(\pi^{\frac{2}{-2+\alpha }}\left(-1+\alpha \right)-S_{0}^{\frac{2}{-2+\alpha }}C_{2}\right)}{\pi^{-\frac{2}{2-\alpha }}\left(-1+\alpha \right)-S_{0}^{\frac{2}{-2+\alpha }}\left(-3+\alpha \right)C_{2}}. \end{array}$$

According to these expressions, we plot the variation of BH temperature (upper graph) and heat capacity (lower graph) with respect to the event horizon radius in Figure 2. From Figure 2 (upper), we observe that the BH temperature is positive only within a specific range of the event horizon radius. For instance, for the parameter values α=0.01 and $C_{2}=15$, the BH temperature is positive when $r_{+}>3.87$; whereas for $r_{+}<3.87$, it corresponds to a physically meaningless negative temperature region. Moreover, the black hole temperature reaches its maximum value when the event horizon radius is $r_{+}\approx 6.73$. The lower graph of Figure 2 shows that the black hole undergoes a phase transition in the framework of GBD modified gravity theory, and the location of the phase transition depends on the model parameter values. For example, for α=0.01 and $C_{2}=15$, the black hole is in an unstable phase with negative heat capacity in the range $0<r_{+}<3.87$, and the heat capacity of BH has its minimum value at the event horizon radius $r_{+}\approx 2.87$. On the other hand, for the range r>3.87, the black hole is in a stable phase with positive heat capacity. Clearly, the black hole undergoes a phase transition at $r_{+}\approx 3.87$, where the BH heat capacity C=0.

By combining Equations (Equation 18), (Equation 25) and (Equation 26), we can derive the corrected entropy expression for black holes under GBD theory, given by:(27)$$\begin{array}{c} S=S_{0}+\frac{{\left(r_{+}^{2-\alpha }\right)}^{\frac{\alpha }{2-\alpha }}\gamma {\left[-1+{\left(r_{+}^{2-\alpha }\right)}^{\frac{2}{-2+\alpha }}C_{2}\right]}^{3}}{4\pi \left(-2+\alpha \right)[1-\alpha +{\left(r_{+}^{2-\alpha }\right)}^{\frac{2}{-2+\alpha }}\left(-3+\alpha \right)C_{2}]}. \end{array}$$

Taking the parameters γ=1, α=0.01 and $C_{2}=15$ into account, we can use Equation (Equation 27) to plot the corrected entropy as a function of the event horizon radius (as shown in Figure 3). From Figure 3, we observe that the entropy *S* with the correction term decreases rapidly with increasing event horizon radius and reaches its minimum value S≈5.44 at $r_{+}\approx 1.09$, then gradually increases. The plot of $S_{0}$ indicates that it increases monotonically with $r_{+}$ increasing. Comparing the two plots of *S* and $S_{0}$, we find that the correction term has a significant effect mainly in the region of small values $r_{+}$, and the entropy-increasing effect is very prominent. As for large values $r_{+}$, the effect of the correction term can be almost ignored (the two curves almost overlap). In addition, we also calculate the influence of other model parameter values on the entropy. From Figure 3 (left), we can see that the variation of the model parameter $C_{2}$ has little effect on the corrected entropy when $r_{+}$ is large.

## 3. Stability Analysis of Particles’ Circular Orbits around a Black Hole under GBD Theory

Some works on the circular orbits of particles can be seen in Refs. [36,63,64,65,66,67], e.g., the issues on the spherical photon orbits around a Kerr BH [63], the dynamics of charged particles moving around Kerr BH with inductive charge and external magnetic field [65], the equivalence between two charged black holes in dynamics of orbits outside the event horizons [36], and the precessing and periodic orbits around hairy black holes in Horndeskis theory [67], were discussed. These studies can be utilized to distinguish different BHs, investigate the properties of spacetime in strong gravitational fields, and subsequently test theories of gravitational interaction. In this section, we analyze the motion and properties of particles around a black hole within the framework of GBD theory. The motion of a free particle in a gravitational field is described by the geodesic equation:(28)$$\begin{array}{c} \frac{d^{2}x^{\mu }}{dp^{2}}+\Gamma_{\nu \lambda }^{\mu }\frac{dx^{\nu }}{dp}\frac{dx^{\lambda }}{dp}=0. \end{array}$$

Here, *p* represents the orbital parameter. The component forms of geodesic equation can be written as
(29)$$\begin{array}{c} \frac{d^{2}t}{dp^{2}}+\frac{B^{{}^{\prime }}}{B}\frac{dt}{dp}\frac{dr}{dp}=0. \end{array}$$
(30)$$\begin{array}{c} \frac{d^{2}r}{dp^{2}}+\frac{BB^{{}^{\prime }}}{2}{\left(\frac{dt}{dp}\right)}^{2}-\frac{1}{2B}{\left(\frac{dr}{dp}\right)}^{2}-rB{\left(\frac{d\theta }{dp}\right)}^{2}-rBsin^{2}\theta {\left(\frac{d\varphi }{dp}\right)}^{2}=0. \end{array}$$
(31)$$\begin{array}{c} \frac{d^{2}\theta }{dp^{2}}+\frac{2}{r}\frac{d\theta }{dp}\frac{dr}{dp}-sin\theta cos\theta {\left(\frac{d\varphi }{dp}\right)}^{2}=0. \end{array}$$
(32)$$\begin{array}{c} \frac{d^{2}\varphi }{dp^{2}}+\frac{2}{r}\frac{d\varphi }{dp}\frac{dr}{dp}+2cot\theta \frac{d\varphi }{dp}\frac{d\theta }{dp}=0. \end{array}$$
where the prime represents the derivative with respect to the radial coordinate *r*. For a gravitational field with spherical symmetry, without loss of generality, we select the initial position and velocity of the particle to be on the equatorial plane, i.e., $\theta =\frac{\pi }{2}$ and $\frac{d\theta }{dp}=0$. Using Equation (Equation 31), we obtain $\frac{d^{2}\theta }{dp^{2}}=0$, which indicates that the particle motion will always remain on the equatorial plane. For particles with non-zero rest mass, the orbital parameter is taken to be the proper time τ. Further derivation leads to two equations for the motion of particles in a gravitational field:(33)$$\begin{array}{c} \frac{dt}{d\tau }=\frac{E}{B}. \end{array}$$
(34)$$\begin{array}{c} r^{2}\frac{d\varphi }{d\tau }=J. \end{array}$$

Here, *E* and *J* are two constants of integration, representing the energy and angular momentum of a unit mass particle. Additionally, using the normalization condition of the four-velocity $g_{\mu \nu }U^{\mu }U^{\nu }=\epsilon$ and Equations (Equation 33) and (Equation 34), we obtain:(35)$$\begin{array}{c} {\left(\frac{dr}{d\tau }\right)}^{2}=E^{2}-B\left(r\right)\left(-\epsilon +\frac{J^{2}}{r^{2}}\right), \end{array}$$
where ϵ=−1 denotes the case of a massive particle, while ϵ=0 corresponds to the case of a massless particle. We define the effective potential as:(36)$$\begin{array}{c} V_{eff}\left(r\right)\equiv B\left(r\right)\left(-\epsilon +\frac{J^{2}}{r^{2}}\right). \end{array}$$

Then Equation (Equation 35) becomes:(37)$$\begin{array}{c} {\left(\frac{dr}{d\tau }\right)}^{2}=E^{2}-V_{eff}\left(r\right). \end{array}$$

Clearly, the properties of the gravitational potential in the GBD theory can be described by Equation (Equation 36), which depends on the relative position and angular momentum of the particles. Obviously, for the case of massive particles, we see that when the radius is r→∞, the effective potential is $V_{eff}=1$ by combining Equations (Equation 5) and (Equation 36). Furthermore, Equation (Equation 37) indicates that when $E^{2}=V_{eff}$, i.e., $\frac{dr}{d\tau }=0$, the orbital radius *r* is constant, and the trajectory of the particle is circular.

The effective potential is crucial to particle radial motions, since the local minimal and maximal values of the effective potential correspond to stable and unstable circular orbits, respectively [26,36,37,68,69,70]. Considering
(38)$$\begin{array}{c} \frac{dV_{eff}}{dr}=0, \end{array}$$
(39)$$\begin{array}{c} \frac{d^{2}V_{eff}}{dr^{2}}=0, \end{array}$$
one can derive the innermost stable circular orbit (ISCO). In black hole physics, it is meaningful to discuss the ISCO, as it is not only the inner boundary but also the starting position of electromagnetic radiation of the accretion disk in the Novikov–Thorne model [37,71,72]. For massless particles (e.g., photon) in our GBD theory, Equation (Equation 38) provides $\left(-2r^{2}+3rC_{1}+4C_{2}\right)/r^{5}=0$, which gives the location of the photon sphere: $r_{ph}=\frac{1}{4}\left(-3C_{1}\pm \sqrt{9C_{1}^{2}-32C_{2}}\right)$. Obviously, for $|C_{1}|\;>\frac{4\sqrt{2}}{3}\sqrt{C_{2}}$ there are two photon spheres, while for $|C_{1}|\;=\frac{4\sqrt{2}}{3}\sqrt{C_{2}}$, there is one photon sphere. Equation (Equation 39) gives the location of the ISCO for photon: $r_{ISCO-ph}=-C_{1}\pm \sqrt{C_{1}^{2}-\frac{10C_{2}}{3}}$.

For the GBD theory, we derive that the relationship that the ISCO with the massive particles needs to satisfy:(40)$$\begin{array}{c} \frac{3r^{2}C_{1}^{2}+8C_{2}^{2}+C_{1}\left(r^{3}+9rC_{2}\right)}{r^{2}\left(rC_{1}+2C_{2}\right)}=0. \end{array}$$

Solving the above equation numerically, for the case of massive particles we can plot the variation of the ISCO radius with respect to the parameters (as shown in Figure 4). For this case, our calculations show that the radius of the ISCO in the GBD modified gravity theory framework will continuously increase with increasing values of the parameters $C_{1}$ and $C_{2}$.

We also investigate the properties of angular momentum of massive particles in the GBD theory. Considering the condition of circular orbits in the equatorial plane:(41)$$\begin{array}{c} \theta =\frac{\pi }{2},\;\frac{d\theta }{d\tau }=0,\;\frac{dr}{d\tau }=0, \end{array}$$
we obtain the expressions for *t* and φ by calculating the components of the motion Equation (Equation 28):(42)$$\begin{array}{c} {\left(\frac{d\varphi }{d\tau }\right)}^{2}=\frac{B^{\prime }\left(r\right)}{r[2B\left(r\right)-rB^{\prime }\left(r\right)]} \end{array}$$
(43)$$\begin{array}{c} {\left(\frac{dt}{d\tau }\right)}^{2}=\frac{2}{B\left(r\right)-rB^{\prime }\left(r\right)}. \end{array}$$

From Equations (Equation 42) and (Equation 43), we derive the expression for angular velocity as:(44)$$\begin{array}{c} \Omega =\frac{\dot{\varphi }}{\dot{t}}=\sqrt{\frac{\partial_{r}B\left(r\right)}{2r}}. \end{array}$$

Furthermore, substituting Equation (Equation 5) into Equation (Equation 44) yields
(45)$$\begin{array}{c} \Omega =\sqrt{\frac{r^{3}-C_{1}r-2C_{2}}{2r^{4}}}. \end{array}$$

Using Equation (Equation 45), we plot the picture of angular velocity as a function of the radius *r* in Figure 5, where parameter values $C_{1}=-10$ and $C_{2}=15$ are selected (corresponding to the two thin solid lines in Figure 5). For this case we can see that for these parameter values, the angular velocity sharply increases in the range of radius *r*: 2.09–3.05, reaches its maximum value Ω≈0.41 at radius r≈3.05, and then Ω decreases to a steady state. In addition, to discuss the variation of angular velocity with respect to the parameters $C_{1}$ and $C_{2}$, we also plot the curves for different values of these parameters in Figure 5. For example, in the left panel of Figure 5, we consider $C_{2}=15$, $C_{1}$ with values of −8, −10, and −12, respectively. In the right panel of Figure 5, we consider $C_{1}=-10$, $C_{2}$ with values of 10, 15, and 20, respectively. From Figure 5, we observe that the values of angular velocity decrease with increasing values of $C_{1}$ or $C_{2}$.

In addition, we can derive to obtain the specific expressions of energy and angular momentum of unit mass particles in GBD modified theory as follows:(46)$$\begin{array}{c} E=\frac{r^{2}+C_{1}r+C_{2}}{r^{2}}\sqrt{\frac{2r^{2}}{2r^{2}+3C_{1}r+4C_{2}}} \end{array}$$
(47)$$\begin{array}{c} J=\sqrt{-\frac{r^{2}\left(C_{1}r+2C_{2}\right)}{2r^{2}+3C_{1}r+4C_{2}}} \end{array}$$

Figure 6 shows the variation of energy and angular momentum of a unit mass particle with radius. From the figure, it can be seen that when the parameter values are taken as $C_{1}=-10$ and $C_{2}=15$, the energy sharply decreases in the radius interval 12.62–24.86, and reaches its minimum value E≈0.93 at the radius r≈24.86. It then gradually increases. The variation trend of angular momentum is similar to that of energy, also reaching its minimum value J≈15.67 at the radius r≈24.86. The variation of *E* and *J* for different parameter values $C_{1}$ and $C_{2}$ can be seen in detail in Figure 6. The plots on the left take the value of $C_{2}=15$, while the plots on the right take the value of $C_{1}=-10$.

To study the issue of circular orbit stability of massive particles moving around a spherically symmetric black hole in GBD theory, we consider the application of the geodesic deviation equation. The geodesic deviation equation is given by:(48)$$\begin{array}{c} \frac{d^{2}\xi^{\alpha }}{d\tau^{2}}+2\Gamma_{\mu \nu }^{\alpha }\frac{dx^{\mu }}{d\tau }\frac{d\xi^{\nu }}{d\tau }+\frac{dx^{\mu }}{d\tau }\frac{dx^{\nu }}{d\tau }\xi^{\beta }\partial_{\beta }\Gamma_{\mu \nu }^{\beta }=0, \end{array}$$
where $\xi^{\alpha }$ is the deviation four-vector. By substituting the spherically symmetric line element (Equation 4) and using Equations (Equation 41)–(Equation 43), we can derive the expressions for the geodesic deviation equation components:(49)$$\begin{array}{c} \frac{d^{2}\xi^{0}}{d\varphi^{2}}+\frac{B^{\prime }}{B}\frac{dt}{d\varphi }\frac{d\xi^{1}}{d\varphi }=0 \end{array}$$
(50)$$\begin{array}{c} \frac{d^{2}\xi^{1}}{d\varphi^{2}}+BB^{\prime }\frac{dt}{d\varphi }\frac{d\xi^{0}}{d\varphi }-2rB\frac{d\xi^{3}}{d\varphi }+\left[\frac{1}{2}{\left(\frac{dt}{d\varphi }\right)}^{2}\left(B^{\prime^{2}}+BB^{\prime \prime }\right)-\left(B+rB^{\prime }\right)\right]\xi^{1}=0 \end{array}$$
(51)$$\begin{array}{c} \frac{d^{2}\xi^{2}}{d\varphi^{2}}+\xi^{2}=0 \end{array}$$
(52)$$\begin{array}{c} \frac{d^{2}\xi^{3}}{d\varphi^{2}}+\frac{2}{r}\frac{d\xi^{1}}{d\varphi }=0 \end{array}$$

Obviously, the solution to Equation (Equation 51) can be expressed as $\xi^{2}=\zeta^{2}e^{i\varphi }$. This indicates that the circular orbits of test particles initially in the equatorial plane will undergo harmonic vibration under perturbations. Therefore, the circular orbit of particle motion is stable. For other equations, assuming the solution takes the form:(53)$$\begin{array}{c} \xi^{0}=\zeta^{0}e^{i\omega \varphi } \end{array}$$
(54)$$\begin{array}{c} \xi^{1}=\zeta^{1}e^{i\omega \varphi } \end{array}$$
(55)$$\begin{array}{c} \xi^{3}=\zeta^{3}e^{i\omega \varphi } \end{array}$$
then substituting Equations (Equation 53)–(Equation 55) into Equations (Equation 49), (Equation 50) and (Equation 52), and considering the stability requirement for circular orbit motion, we derive the following constraint:(56)$$\begin{array}{c} \omega^{2}=3B-2rB^{\prime }+\frac{rBB^{\prime \prime }}{B^{\prime }}\geq 0. \end{array}$$

This constraint (Equation 56) can also be obtained by using Equation (Equation 39) and the consideration $\frac{d^{2}V_{eff}}{dr^{2}}\geq 0$. By substituting expression (Equation 5) into the above Equation (Equation 56), we show in Figure 7 the dependence of the parameter $\omega^{2}$ on the circular orbit radius *r*. When the integral constants are taken as $C_{1}=-10$ and $C_{2}=15$, we find that the stable region of circular orbits $\omega^{2}\geq 0$ is: r≳24.86. The effects of other model parameter values on the parameter $\omega^{2}$ and the corresponding stable circular orbit regions are shown in Figure 6 (the left figure with $C_{2}$ taken as 15 and the right figure with $C_{1}$ taken as −10).

## 4. Conclusions

The emergence of challenging problems such as gravity quantization and the origin of dark matter and dark energy has provided motivation for finding gravity theories beyond Einstein’s general relativity. Researchers in the field of gravity have made many efforts and practices in exploring modified or extended theories of general relativity, and the study of applying modified gravity theories to astrophysics and cosmology has increasingly received people’s attention. This article focuses on the properties of a BH solution from the so-called GBD theory. We investigate the thermodynamic corrected-entropy problem of static spherically symmetric BHs and the stability of particles’ circular orbits around a BH under the framework of the GBD modified gravity theory. Since studying the structure of geodesics in strong gravitational fields plays an important role in exploring the physical characteristics of compact objects, we firstly quantitatively analyze the contribution of the corrected term to the entropy in the GBD gravity. It is found that the effect of the correction term on entropy is significant when the value of the event horizon radius $r_{+}$ is small, while its effect can be negligible when $r_{+}$ is large. Secondly, the article also analyzes and shows the dependence relationship of black hole mass, temperature, heat capacity, etc., on the event horizon radius. Based on the motion equations, we investigate the relevant properties of physical quantities such as energy, gravitational potential energy, angular velocity, and angular momentum of test particles under the GBD theory. Finally, the article applies the geodesic deviation equation to analyze the stable circular orbit of test particles in the GBD theory and finds that the particle undergoes harmonic motion in the equatorial plane, and the range of the restricted area that the radial coordinate needs to satisfy to achieve stable circular orbit motion is given.

## Figures and Tables

**Figure 1 entropy-25-00814-f001:**
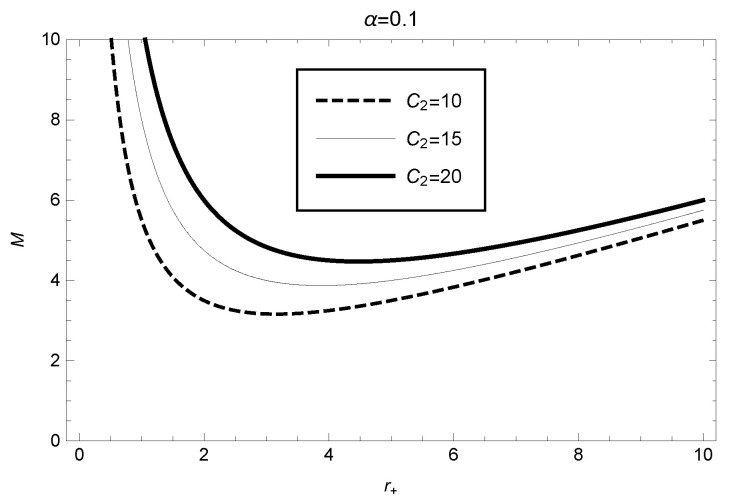
Taking the different values of $C_{2}$, the mass for the spherically symmetric black hole as a function of the event horizon radius in the framework of GBD modified theory.

**Figure 2 entropy-25-00814-f002:**
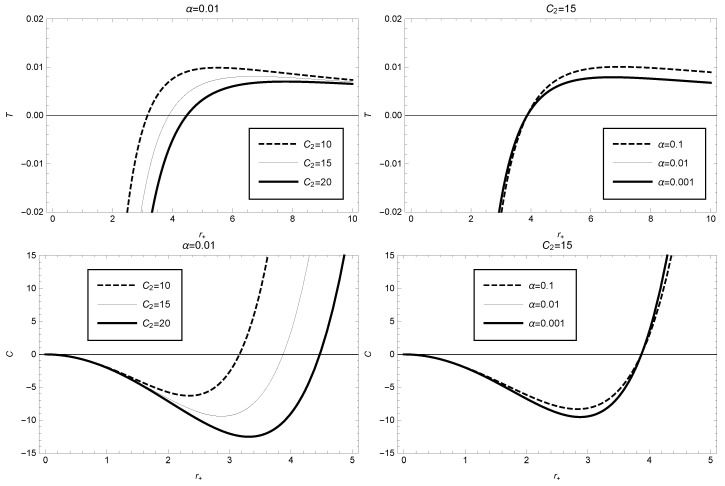
Black hole temperature (**upper**) and heat capacity (**lower**) as function of the event horizon radius in GBD theory, where α=0.01 (or $C_{2}=15$) has been taken in the **left** (or **right**) figure.

**Figure 3 entropy-25-00814-f003:**
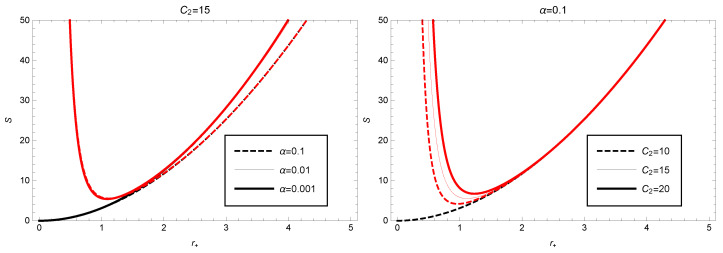
Black hole thermodynamic correction entropy as function of the event horizon radius in GBD theory (the red curve denotes the variation of *S*, and black curve denotes the variation of $S_{0}$), where $C_{2}=15$ (or α=0.01) has been taken in the **left** (or **right**) figure.

**Figure 4 entropy-25-00814-f004:**
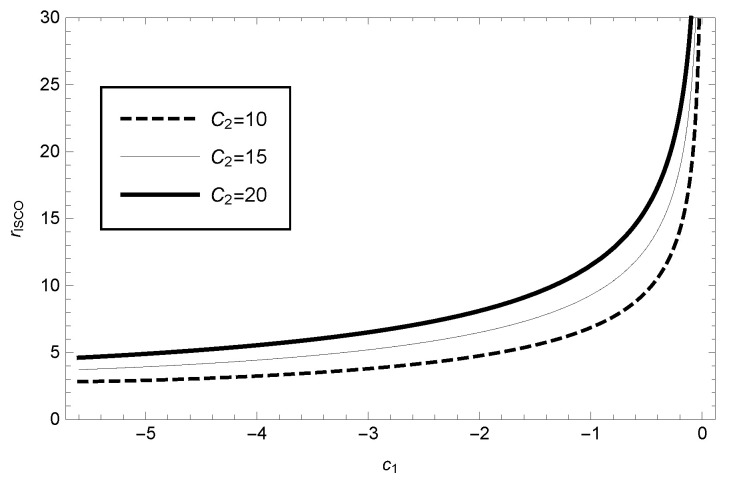
The radius of the ISCO as function of the parameter $C_{1}$ in GBD theory for the case of massive particles, where the different values of $C_{2}$ have been taken.

**Figure 5 entropy-25-00814-f005:**
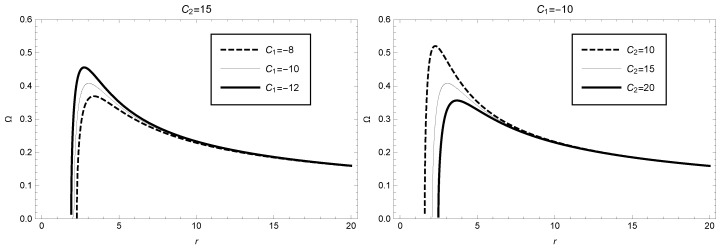
Angular velocity as function of the radius *r* in the framework of GBD modified theory, where $C_{2}=15$ (or $C_{1}=-10$) has been taken in the **left** (or **right**) figure.

**Figure 6 entropy-25-00814-f006:**
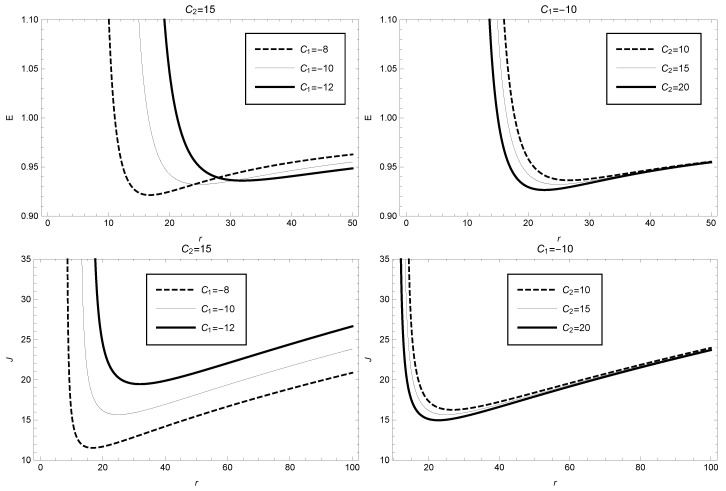
Specific energy (**upper**) and angular momentum (**lower**) as function of the radius in GBD theory, where $C_{2}=15$ (or $C_{1}=-10$) has been taken in the **left** (or **right**) figure.

**Figure 7 entropy-25-00814-f007:**
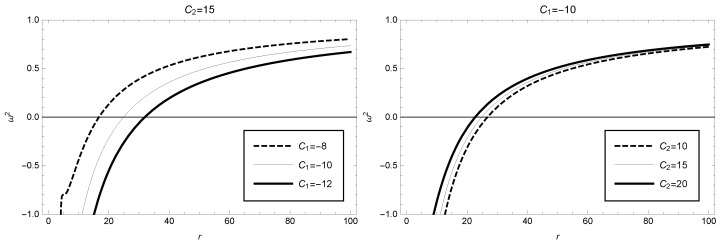
$\omega^{2}$ as function of the radius in the framework of GBD modified theory, where $C_{2}=15$ (or $C_{1}=-10$) has been taken in the **left** (or **right**) figure.

## Data Availability

Not applicable.

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
