# Peer review of "Properties of Spherically Symmetric Black Holes in the Generalized Brans–Dicke Modified Gravitational Theory"

_entropy, 2023, doi:10.3390/e25050814_

Round 1

Reviewer 1 Report

See my comments and suggestions at the attached file: Report_Entropy.docx

 Minor editing of English language required

Reviewer 2 Report

This manuscript focused on properties of spherically symmetric black hole in the generalized Brans-Dicke modified gravitational theory, such as the contribution of the corrected term to the entropy, the dependence of black hole mass, temperature and heat capacity on the event horizon radius, and stable circular orbit of test particles. It is interesting. However, there are many unclear or incomplete presentations. If the following comments and suggestions are considered fully, this paper may be recommended for publication in the Entropy journal.

Query 1: The introduction of modified gravitational theories is incomplete and lacks some important references. Besides the importance of modified gravitational theories, several important modified gravitational theories, e.g., scalar-tensor theories (Int. J. Theor. Phys. 1968, 1, 25–36; Sci. China Phys. Mech. Astron. 2011, 54, 2071–2077; Phys. Rev. D 2012, 86, 044007; Sci. China Phys. Mech. Astron. 2015, 58, 030002; Phys. Rev. D 2016, 93, 044013; Phys. Rev. D 2021, 103, 064040) and scalar–tensor–vector gravity (J. Cosmol. Astropart. Phys. 2006, 3, 4; Phys. Rev. D 2009, 79, 044014.) should be introduced. The authors should show the characteristics, differences and roles of these modified gravitational theories.

Query 2: This manuscript focused on properties of spherically symmetric black hole in the generalized Brans-Dicke modified gravitational theory. Why were other properties not introduced? Some dynamical and physical properties of black holes in modified theories of gravity should be described appropriately [see e.g. Universe, 7, 488 (2021); Universe, 8, 320 (2022); General Relativity and Gravitation 54:110 (2022); Eur. Phys. J. C 82: 885 (2022); Eur. Phys. J. C (2023) 83:264; The Astrophysical Journal, 940: 166, 2022].

Query 3: What do C_1 and C_2 in Eq. (5) mean from a viewpoint of physics?

Query 4: The authors should compare the properties of the spacetime metric (4) with those of the regular black hole and Reissner–Nordström (RN) black hole [The Astrophysical Journal,2021, 909, 22; The Astrophysical Journal Supplement Series,2021, 254, 8; General Relativity and Gravitation 54:110 (2022)]

Query 5: Why were the circular orbits of particles studied in astrophysics? The authors should introduce some latest works on this topic. For example, Gen. Relativ. Gravit. 35, 1909 (2003); Gen. Relativ. Gravit. 53, 10 (2021); Universe, 7, 410 (2021); Phys. Rev. D 105,124039 (2022); General Relativity and Gravitation 54:110 (2022); Eur. Phys. J. C 83: 311 (2023)

Query 6: Why were the circular orbits of photons not studied? In fact, the circular photon orbits are closely related black hole shadows. Please see the latest works of black hole shadows, e.g., Eur. Phys. J. C 82: 831 (2022), General Relativity and Gravitation 54:93 (2022), Eur. Phys. J. C (2022) 82: 885, General Relativity and Gravitation (2022) 54:110, Symmetry 14, 2237 (2022), and Eur. Phys. J. C (2023) 83:264. Even if the circular orbits of photons are not studied, these works on the black hole shadows should be introduced.

Query 7: line 83 of page 3: C_2 ≤0 should be C_2≥0.

Query 8: The title of Section 3 is “Analysis of geodesic motion stability of particles in black holes”. This is not true. The geodesic motion of particles in black holes is stable or unstable. In fact, the authors considered only the stability of particles’ circular orbits.

Query 9: The stability of particles’ circular orbits can be discussed simply by using Eq. 39 with the sign “=” replaced by “≥”. Please check whether this path leads to the obtainment of Eq. (56). If it does, the authors should show this point.

Round 2

Reviewer 2 Report

The authors answered my queries and improved their paper according to my suggestions. Now, I would like to recommend the present version of the paper for publication in this journal. However, some minor corrections and revisions are still necessary before a final version of the manuscript is sent to the Editorial Office.

Point 1: In the title of the paper, it is good to replace “framework of GBD” with “generalized Brans-Dicke”.

Point 2: The presentations in language should be improved further. Some examples are given as follows.

Line 60: It is good to insert “to” between “related” and “black”.

Line 289: It is good to write “can also” instead of “also can”.

Author Response

List of Changes for Manuscript ID:Entropy-2381628

Dear Editor,

Thanks for your correspondence. According to the honorable referees' comments on the manuscript entitled "Properties of spherically symmetric black hole in the framework of GBD modified gravitational theory" (Manuscript ID:Entropy-2381628), we have revised our manuscript. The introduction of changes is as follows.

Reply to the Review Report (Reviewer 2)

Point 1: In the title of the paper, it is good to replace “framework of GBD” with “generalized Brans-Dicke”.

Reply 1: According to the referee’s instruction, we replaced “framework of GBD” with “generalized Brans-Dicke” in the title of the revised manuscript.

Point 2: The presentations in language should be improved further. Some examples are given as follows.

Line 60: It is good to insert “to” between “related” and “black”.

Line 289: It is good to write “can also” instead of “also can”.   

Reply 2: According to the referee’s comment, we try our best to improve the presentations in language for this manuscript. For example, we inserted “to” between “related” and “black” in Line 60, and wrote “can also” instead of “also can” in Line 289, and others (please see the revised manuscript).

We thank the anonymous referee for his/her very instructive comments, which improve our paper greatly.

Thanks a lot.

Yours sincerely

Jianbo Lu

Department of Physics, Liaoning Normal University